# MicroRNAs as Potential Biomarkers in Gynecological Cancers

**DOI:** 10.3390/biomedicines11061704

**Published:** 2023-06-13

**Authors:** Joanna Miśkiewicz, Aleksandra Mielczarek-Palacz, Joanna Magdalena Gola

**Affiliations:** 1Department of Immunology and Serology, Faculty of Pharmaceutical Sciences in Sosnowiec, Medical University of Silesia in Katowice, 41-200 Sosnowiec, Poland; joanna.miskiewicz@sum.edu.pl (J.M.); apalacz@sum.edu.pl (A.M.-P.); 2Department of Molecular Biology, Faculty of Pharmaceutical Sciences in Sosnowiec, Medical University of Silesia in Katowice, 41-200 Sosnowiec, Poland

**Keywords:** miRNA, gynecological cancers, biomarkers

## Abstract

MicroRNAs are non-coding transcripts that, thanks to the ability to regulate the mRNA of target genes, can affect the expression of genes encoding tumor suppressors and oncogenes. They can control many important cellular processes, including apoptosis, differentiation, growth, division, and metabolism. Therefore, miRNAs play an important role in the development of many cancers, including gynecological cancers. Ovarian cancer, endometrial cancer, cervical cancer, and vulvar cancer are the most common cancers in women and are a frequent cause of death. The heterogeneity of the pathogenesis of these gynecological diseases makes the diagnostic process a significant obstacle for modern medicine. To date, many studies have been carried out, in which particular attention has been paid to the molecular pathomechanism of these diseases, with particular emphasis on miRNAs. To date, the changed profile of many miRNAs, which influenced the promotion of proliferation, migration, invasion processes and the simultaneous inhibition of programmed cell death, has been proven many times. Detailed understanding of the molecular effects of miRNAs in the above-mentioned gynecological cancers will enable the development of potential predictive and prognostic biomarkers, as well as the optimization of the diagnostic process.

## 1. Introduction

MicroRNAs (miRNAs) are a group of small, non-coding, approximately 22 nucleotide RNAs that regulate the expression of many genes [1,2]. MicroRNAs participate in many cellular processes, including embryogenesis, organogenesis, metabolism, apoptosis, and cell differentiation [3,4]. To date, ongoing research on the roles of miRNAs has shown that they can affect pathogenesis, predictive factors, and therapy in almost all cancers, including gynecological cancers. Gynecological cancers are the most common cancers in women. They may be e.g., in the ovaries, endometrium, cervix and vulva. The still ambiguous pathogenesis, heterogeneity of diseases and difficult diagnostics mean that gynecological cancers are often diagnosed at a late stage. Therefore, research is still underway on new, sensitive, and specific biomarkers to enable quick, non-invasive diagnostics in the early stages of the disease, prediction, and precise monitoring of therapy [5,6,7,8]. MicroRNAs, which have been studied for many years, provide hope. Several studies have shown that the expression level of individual miRNAs may be a new, sensitive, and specific biomarker [9].

In this paper, the relationship between miRNAs and ovarian cancer, endometrial cancer, cervical cancer, and vulvar cancer will be discussed.

## 2. MicroRNA Particles

The biogenesis of miRNAs is subject to complex regulation and even minor disturbances in miRNA biogenesis may negatively affect their expression, as well as the ability to regulate target genes. The process of miRNA biogenesis involves two pathways: canonical and non-canonical [1].

### 2.1. Canonical Biogenesis Pathway

MicroRNAs are encoded as clusters, containing several to several hundred miRNAs, or as single genes [10]. MicroRNA biogenesis begins with transcription by RNA polymerases II and III, followed by further processing of the resulting transcripts [4,11]. The dominant pathway of miRNA biogenesis is the canonical pathway, which includes two stages of maturation. In the first stage of the canonical pathway, which runs in the cell nucleus, pri-miRNA is converted into pre-miRNA by the so-called a microprocessor complex composed of the Drosha enzyme (ribonuclease III enzyme) that cuts the pri-miRNA duplex at the base of the hairpin structure, which cooperates with the RNA-binding protein-DGCR8 (DiGeorge Syndrome Critical Region 8), recognizing pri-miRNA motifs, e.g., N6-methyladenylated GGAC [11,12,13,14]. After the formation of the pre-miRNA, thanks to the participation of exportin 5/RanGTP, it is transported to the cytoplasm where it is processed by the Dicer enzyme belonging to ribonucleases III [11,12,15]. This endoribonuclease consists of two RNAse III catalytic domains, a PAZ domain, a double-stranded RNA-binding domain, and an RNA/ATPase helicase domain [1,14,15,16,17]. Complex Dicer-TRBP (TAR-RNA binding protein) processing involves the removal of the terminal loop, which leads to the formation of a mature miRNA [1,18,19]. Depending on the directionality of the miRNA strand, a 3p miRNA strand can be obtained if it comes from the 3’ end of the pre-miRNA and a 5p strand if it comes from the 5’ end [11,20]. In addition, the leading and passenger threads are distinguished. The leading strand is inserted to the RISC complex (RNA-induced silencing complex), which plays the role of silencing the expression of target genes. The RISC complex consists of i.a. the Argonaute protein family members (AGO 1–4). The miRNA directs the RISC complex to complementary mRNA target sequences. The miRNA interacts with the 3’UTR region of the mRNAs [1,20,21,22,23]. The passenger thread usually degrades [11,20].

### 2.2. Non-Canonical Biogenesis Pathway

The second pathway of miRNA biogenesis is the non-canonical pathway, using the canonical pathway proteins: Dicer enzyme, Exportin 5, Drosha enzyme and AGO2. The non-canonical pathway can be divided into the Dicer enzyme-independent pathway and the Drosha/DGCR8 enzyme-independent pathway [11,24,25]. In the case of the Dicer-independent pathway, the Drosha enzyme processes short hairpin RNA transcripts—shRNA [11,26]. The maturation of miRNA ends with attachment to the AGO2 protein and cleavage of the 3’ and 5’ strands [11,27]. An example of substrate of a pathway independent of the Drosha/DGCR8 enzyme is transcript with a 7-methylguanosine cap and mirtrons [11,24,25,28].

### 2.3. Regulation of Gene Expression Via miRNAs

The mechanism of miRNA-mediated regulation of target gene expression is complex. The miRNA binds to the 3’UTR and 5’UTR of the target mRNA, thereby repressing translation in the case of the 3’UTR and silencing the expression of target genes in the case of the 5’UTR. In addition, the miRNA has the ability to bind within the promoter region and the coding sequence. Attachment within the coding sequence has a silencing effect on gene expression [11,29,30,31,32,33]. In contrast to the previously mentioned mechanisms, the attachment of miRNA to the promoter induces transcription [11,34].

The regulation of target gene expression is also mediated by the miRISC complex consisting of the miRNA leader/guide strand and the AGO protein. The miRISC complex is able to bind to the target mRNA. Full complementarity between mRNA and miRNA results in activation of AGO2 and induces mRNA degradation [11,35,36,37]. MicroRNAs may negatively affect the mRNA of genes encoding oncogenes and tumor suppressors. Thus, miRNAs can play the role of tumor suppressors or oncogenes [38,39].

## 3. Role of miRNAs in Ovarian Cancer

Ovarian cancer (OC) is a malignant disease that is a common cause of death among women [40]. Ovarian cancer is classified according to the FIGO classification common to ovarian, fallopian tube, and peritoneal cancer [41]. Due to the heterogeneous nature of OC, the pathogenesis of the cancer has not yet been clearly defined. However, genetic factors, especially BRCA1 and BRCA2 gene mutations, are indicated as the main cause of the disease [42,43,44,45]. In 2004, the theory of the so-called “two paths” divided ovarian cancer into: low grade type I (mucinous, serous, clear cell, endometrioid, and transitional histological types) and high-grade type II (serous, carcinosarcoma, and undifferentiated). It was found that type I ovarian cancers include lesions originating directly from the ovary [46,47]. Mutations present in type I ovarian cancer activate the MAPK pathway (mitogen-activated protein kinases) through *BRAF* (rapidly accelerated fibrosarcoma B-type) and *KRAS* (Kirsten rat sarcoma viral oncogene homology) mutations [46,47,48]. In contrast, type II ovarian cancer lesions originate outside the organ, including in the fallopian tube. Moreover, type II cancer is genetically unstable, aggressive and is characterized by increased expression of *HER2/neu* and *AKT* genes as well as mutations of *TP53* and *BRCA1, BRCA2* genes [46,47,49]. Attention is also paid to miRNAs that play an important role in the pathogenesis of ovarian cancer. To date, many studies have been conducted showing that miRNAs affecting the processes of proliferation, invasion, apoptosis, and control of the cell cycle have an altered profile [40,50].

Analyzing formalin-fixed tissues, significant amounts of miRNAs with altered expression were found in cancerous tissue compared to healthy tissue.: Braicu et al. [51] showed increased expression of miR-93, miR-200c, miR-141, miR-492, miR-429, miR-155, miR-205, miR-200a, miR-200b versus normal tissue [51]. As indicated in other studies [52], increased expression of miR-200a, miR-200b, and miR-200c may be associated with poor prognosis in patients with epithelial ovarian cancer. Therefore, these three miRNAs may be biomarkers in OC [52]. These miRNAs, together with miR-141 and miR-429, belong to the miR-200 family and play a significant role in epithelial-mesenchymal transition (EMT) negative regulation by targeting mRNAs of key regulators of this process, e.g., ZEB1 (Zinc Finger E-Box Binding Homeobox 1), ZEB2 (Zinc Finger E-Box Binding Homeobox 2), TGFBR1 (Transforming Growth Factor Beta Receptor 1), SMAD2 (SMAD Family Member 2) or c-Myb [53].

Braicu et al. [51] showed that an important prognostic indicator may be miR-182 [51], which is involved in the initiation of migration, proliferation, and invasion processes [54]. In addition, there are a few reports explaining the functions of miR-492 and miR-325 and they apply to other types of cancer. The indicator MiR-492, described in breast cancer, promotes cell proliferation by targeting the suppressor SOX7 [55]. The indicator MiR-325, in bladder cancer, showed low expression associated with reduced patient survival [56]. A study by Braicu et al. [51] also showed deregulation of let-7b family members [51]. The let-7 family members play a special diagnostic, prognostic, and therapeutic role due to the important function of inhibiting oncogenes such as *HRAS*, *KRAS*, *HGMA-2* and *c-MYS54* [51,57,58,59,60].

The profiling of miRNAs in epithelial ovarian cancer tissues has allowed for the identification of molecules that enable the differentiation of cancerous tissues from healthy tissues with 97% sensitivity and 92% specificity, including the increased expression of hsa-miR-182-5p, hsa-miR-96-5p, hsa-miR-183-5p, hsa-miR-182-3p, hsa-miR-15b-5p, hsa-miR-141-5p, hsa-miR-135b-3p, and hsa-miR-130b-5p and the reduced expression of hsa-miR-1271-5p and hsa-miR-574-3p [61]. Furthermore, Marton et al. [62] revealed a set of nine miRNAs with higher expression in people with ovarian cancer compared to healthy people: miR-200a, miR-200c, miR-200b, miR-429, miR-141, miR-34a, miR-34b, miR-203a [62].

As ovarian cancer develops, cancer cells reorganize their stroma, causing fibroblasts to transform into cancer-associated fibroblasts (CAFs), promoting proliferation, invasion, and metastasis [63]. It was shown that in CAF fibroblasts, miR-214 and miR-31 were underexpressed, and miR-155 was overexpressed compared to non-cancerous fibroblasts. Deregulation of these miRNAs may affect the conversion of normal fibroblasts to CAF. The possibility of converting CAF to normal fibroblasts has also been demonstrated [64].

Feng et al. [65] showed that miR-29c-3p may be a prognostic biomarker in patients with ovarian cancer. The authors showed the reduced expression of miR-29c-3p and the increased expression of kinesin *KIF4A* (Kinesin family member 4A) acting as an oncogene. Cross-regulation between miR-29c-3p and *KIF4A* has also been demonstrated. Patients with higher expression of this miRNA showed a higher survival rate. The *KIF4A* oncogene silencing has been shown to inhibit cell proliferation and migration in ovarian cancer. In addition, the expression of *KIF4A* may be influenced by the miR-29c-3p, the increased expression of which inhibits the proliferation and migration of cells in ovarian cancer [65].

Researcher attention has focused on exosomal miRNAs, which may prove to be an alternative for labeling miRNAs in tissues. Taylor et al. [66] showed altered expression of eight exosomal miRNAs: miR-141, miR-21, miR200b, miR-200a, miR-200c, miR-205, miR-203, miR-214. These molecules were expressed similarly in both exosomal and tissue-derived forms. Thus, it was proved that circulating miRNAs reflect the miRNA profile in cancer [66]. Focusing on exosomal miRNAs, it was also shown that the expression of miR-99a-5p was increased in epithelial ovarian cancer [67]. Transfection of human peritoneal mesothelial cells (HPMCs) with miR-99a-5p caused an increase in the expression of fibronectin and vitronectin, which play important roles in the spread of ovarian cancer to the peritoneum [67]. In addition, eight miRNAs were selected, based on which two predictive models of ovarian cancer were created: miR-26a-5p, miR-142-3p, let-7d-5p, miR-374a-5p, miR-766-3p, miR-200a-3p, miR-328-3p, and miR-130b-3p [57]. Models will make it possible to distinguish healthy patients from those with ovarian cancer and will allow to distinguish benign tumors from early-stage cancer [57]. In addition, miRNA profiling studies in the plasma of OC patients were conducted, showing increased expression of miR-200c-3p, miR-21-5p, miR-221-3p, and miR-484, and decreased expression of miR451a and miR-195-5p [68]. It has been shown that, from the diagnostic point of view, determination of the expression of hsa-miR-221-3p together with hsa-miR-200c-3p increases the sensitivity and specificity of OC [68]. In Table 1, the miRNAs with altered expression in OC are shown.

## 4. Role of miRNAs in Uterine Cancer

Uterine cancer is one of the most common gynecological malignancies. The increase in the incidence of uterine cancer is due to the constantly increasing and ageing population, and the obesity rate [70,71,72]. Endometrial cancer is classified according to the FIGO classification [73]. The main symptom of endometrial cancer is prolonged menstrual or intermenstrual bleeding. In addition, bleeding in the postmenopausal period may indicate the presence of endometrial cancer [70,74,75]. There are two types of endometrial cancer: type I associated with the endometrium and estrogen stimulation, and type II not associated with the endometrium. These types are also associated with genetic disorders. In type I, there are the following genetic disorders: a mutation in the suppressor gene *PTEN*, microsatellite instability and mutations in the *PIK3CA* (phosphatidylinositol kinase) gene resulting in the intensification of the cell proliferation process and inhibition of the programmed cell death process, as well as *CTNNB1* mutations affecting the reduction of β-catenin expression and thus the intensification of cells’ invasion [76,77,78,79,80,81]. In type II, there are mutations in the genes encoding p16 and p53 occur most often [76]. In addition, the *HER2* (human epidermal growth factor receptor 2) gene is amplified [70,73]. In endometrial cancer, the following pathways are disturbed: WNT-β catenin, RAS-MERK-ERK and PI3K-PTEN-AKT-mTOR [76,82,83]. In addition, the development of endometrial cancer is associated with single nucleotide polymorphism in the *KLF* (Krueppel-like factor 1), *HNF1B* (hepatocyte nuclear factor 1 homeobox B), *CYP19A1* (cytochrome P450 family 19 subfamily A member 1), *EIF2AK* (eukaryotic translation initiation factor 2 alpha kinase), *MYC* (MYC proto-oncogene), and *SOX4* (SRY-box transcription factor 4) genes [76,84]. In the pathogenesis of endometrial cancer, miRNA also has an important role, which, according to many studies conducted so far, affects the development of this type of gynecological cancer.

MicroRNAs show diagnostic and prognostic potential in endometrial cancer. Research is still underway to identify the relationship between the change in miRNA expression and the pathological factors of endometrial cancer. Lee et al. [85] demonstrated significantly increased expression of miR-183, miR-182, miR-200c, miR-205, and miR-200a in endometrial cancer compared to normal endometrial tissue, complex atypical hyperplasia, and simple hyperplasia [85]. It has been shown that as a result of increased expression of miR-182, the expression of the target gene-*TCEAL7* (transcription elongation factor A-like 7) was reduced in endometrial cancer, which led to the increased expression of c-Myc, cyclin D1 and activation of NFκB. Reducing the expression of the endometrial cancer oncogene miR-182 led to the inhibition of colony formation and cell proliferation as a result of the abolition of the inhibition of the suppressor gene *TCEAL7* [86].

Subsequent studies by Pan et al. [87] showed the influence of the miR-125b-5p/MTFP1 axis on the development of endometrial cancer by affecting cell proliferation, migration, and invasion [87]. The reduced expression of the miR-125b-5p and overexpression of *MTFP1* (mitochondrial fission process 1) in cancer have been shown, so it can be concluded that miR-125b-5p acts as a suppressor and *MTFP1* as an oncogene [87]. Important processes affecting the development of endometrial cancer are also affected by miR-576-5p. Significant overexpression of miR-576-5p was demonstrated to induce proliferation, migration, and invasion of tumor cells. In addition, miR-576-5p overexpression was found to significantly reduce amounts of E-cadherin and increase amounts of N-cadherin, β-catenin, vimentin and ZEB1, facilitating metastasis of cancer cells. It was also shown that the target of miR-576-5p is the *ZBTB4* gene (zinc finger and BTB domain containing 4). Overexpression of miR-576-5p inhibits *ZBTB4* expression. The discussed miRNA regulates the expression of ZBTB4 both at the mRNA and protein levels. Increasing the expression of ZBTB4 inhibited the process of cell proliferation, colony formation and metastasis, while reducing the expression induced cell proliferation, colony formation and metastasis. Thus, it was concluded that miR-576-5p may regulate the development of tumor cells in endometrial cancer by targeting *ZBTB4* [88].

Other identified molecules undergoing aberrant expression are downregulated compared to the healthy endometrium: miR-199a-3p, miR-221-3p, miR-125b-5p miR-23b, miR-451, and miR-146-5p. Moreover, miR-34a-5p has higher expression in estrogen-dependent endometrial cancer that is associated with endometrial hyperplasia compared to expression in estrogen-dependent endometrial cancer that is associated with endometrial atrophy [89]. In addition, the authors of the discussed studies showed that the expression of the miR-23b was associated with poorer survival of patients with endometrial cancer [89].

Endometrial cancer is one of the most threatening gynecological malignancies in women. MicroRNAs offer hope for determining factors that predict overall patient survival. Wang et al. [90] showed the signature of six miRNAs: hsa-mir-142.MIMAT0000433, hsa-mir-15a.MIMAT0000068, hsa-mir-3170.MIMAT0015045, hsa-mir-142.MIMAT0000434, hsa-mir-146a.MIMAT0000449, and hsa-mir-1976.MIMAT0009451 strongly correlated with the overall survival of patients with endometrial cancer [90]. Huang et al. [91] identified four miRNAs that are prognostic markers of patients with endometrial cancer: miR-34a-5p, miR-31-5p, miR-4772-3p, and miR-26a-1-3p [91]. Wu et al. [92] also created a model consisting of four miRNAs: miR-4758, miR-142, miR-876, miR-190b, enabling survival prognostication of patients with endometrial cancer [92]. Moreover, Lu et al. [93] created a prognostic model consisting of 11 miRNAs, including three molecules with reduced expression: hsa-miR-3614, hsa-miR-3170, and hsa-miR-4687, and eight with increased expression: hsa-miR-215, hsa-miR-592, hsa-miR-216b, hsa-miR-1301, hsa-miR-940, hsa-miR-363, hsa-miR-7110, and hsa-miR-96 [93].

The prognostic factor considered critical in patients are lymph nodes, which are the first spread place of cancer cells. Metastasis to pelvic lymph nodes is related to the stage of cancer development [94,95,96,97]. Routine lymphadenectomy is being replaced by sentinel lymph node mapping. Downregulation of miR-204-5p was associated with lymph node metastasis. This molecule has turned out to be an excellent biomarker for the assessment of lymph nodes along with their mapping [94].

Another molecule with an important function in endometrial cancer is miR-29a-3p [98]. Geng et al. [98] showed that the reduced expression of this miRNA was associated with poor prognosis, indicating its suppressor function. Upregulation of miR-29a-3p resulted in inhibition of proliferation, migration, invasion, and colony formation. Vascular endothelial growth factor A (VEGFA), whose expression was inhibited by miR-29a-3p, has been shown to be a direct target of this molecule. In addition, it has been shown that miR-29a-3p may inhibit the development of endometrial cancer by inhibiting the CDC42/PAK1 pathway through VEGFA, which is important in tumor development [98].

The expression profile of individual miRNAs may vary depending on the type of material collected for testing. For example, Chung et al. [99] showed an increase in the expression of miR-200c and miR-200a in endometrial cancer specimens. Furthermore, the study by Hiroki et al. [100] showed an increase in the expression of members of the miR-200 family in endometrial cancer specimens (except for miR-141, which showed no change in the expression). Conversely, Roman-Canal et al. [101] showed a decrease in the expression of all miR-200 family members in extracellular vesicles (EVs) obtained from the peritoneal lavage of EC patients; however, among them, only miR-200b-3p met the biomarker criteria, based on the results of predictive analysis using logistic modeling. The reasons for the discrepancy between the miRNA profiles in cancer tissues and EVs are not fully understood. Roman-Canal et al. [101] postulated that EVs isolated from peritoneal lavage may also originate from other sources, including immune cells, hence the differences in the miRNA profiles. As mentioned above (in part: *MicroRNAs in ovarian cancer*), the miR-200 family members are negative regulators of EMT [53]. However, in endometrial cancer, they can also act as ‘oncomiRNAs’ by targeting mRNAs of tumor suppressors, such as PTEN (Phosphatase and Tensin Homolog) [53].

Table 2 lists the miRNAs found in endometrial cancer that are potential biomarkers.

## 5. Role of miRNAs in Cervical Cancer

Cervical cancer (CC) is another cancer with a high incidence among women. More than 95% of cervical cancer cases are related to human papillomavirus (HPV) infection. Particularly crucial in the etiopathogenesis of CC is the integration of DNA HPV with the human genome. The HPV proteins participate in the development of chronic inflammation, which is an important etiopathogenetic factor of CC [104,105,106,107]. Genetic studies have allowed the identification of changes in signaling pathways such as TGFβ and PI3K/MAPK. In addition, mutations in *KRAS*, *PTEN* and *ARID1A* are common [108]. An important role in the molecular etiopathogenesis of cervical cancer is also played by the APOBEC enzyme (apolipoprotein B mRNA editing enzyme, catalytic polypeptide), affecting the changes of the RNA or DNA sequence through the deamination of cytosine to uracil. It has been shown that APOBEC is induced in cells that have been infected with HPV virus [109,110,111]. The key players are the miRNAs that regulate the expression of many important genes influencing the cervical cancer etiopathogenesis. They influence the initiation of the inflammatory process stimulating the development of CC [106].

One of the molecules showing a relationship between its expression and the clinical characteristics of patients is miR-599, which shows reduced expression in cervical cancer, which is associated with a poor prognosis for patients [112]. The miR-599 expression is associated with FIGO stage and lymph node metastasis. Increasing the expression of miR-599 affects the inhibition of migration, invasion, and proliferation of cancer cells. Reducing the expression of the molecule in question has the opposite effect. These studies indicate that miR-599 may be a potential predictive biomarker [112]. Another molecule with reduced expression in cervical cancer is miR-489. The overexpression of miR-489 is associated with inhibition of proliferation and the induction of apoptosis by regulating the PI3K/AKT/P53 pathway [113].

A significant influence of miRNAs on cell apoptosis in cervical cancer was also found, including, among others, miR-1298-3p. Significant reduction of its expression in CC cell lines was demonstrated. In addition, it has been shown that overexpression of miR-1298-3p inhibits cell proliferation by enhancing the apoptosis process. It was also shown that with an increase in the expression of miR-1298-3p, the pro-apoptotic protein Bax was induced, and the expression of the anti-apoptotic protein Bcl-2 was reduced. Increased expression of miR-1298-3p also inhibits the invasion and migration of cancer cells [114].

The effect of hypoxia on key functions in cervical cancer development and progression, including miRNAs, has also been investigated. One of the factors affecting the transcription of genes responsible for cell adaptation to hypoxia is the hypoxia-inducible factor (HIF). The family of HIF factors includes HIF-1α, HIF-2α, and HIF-3α [115,116]. Gao et al. [115] showed that the increased expression of HIF3α caused by hypoxia increases the expression of the miR-630, and at the same time the increased expression of miR-630 induces overexpression of HIF-3α. Thus, a feedback loop between miR-630 and HIF-3α has been demonstrated. Overexpression of miR-630 causes inhibition of apoptosis, enhanced radioresistance, invasion, metastasis, and migration. Thus, both miR-630 and HIF-3α may be potential biomarkers in cervical cancer [115].

MicroRNAs can be important diagnostic biomarkers. Feng et al. [117] showed that miR-95-3p is an important prognostic biomarker in cervical cancer. According to the research conducted by the authors, overexpression of this molecule is associated with poor prognosis, promotion of migration, and proliferation of cervical cancer cells. It has also been shown that miR-95-3p targets the vascular cell adhesion molecule 1 (VCAM1). The mentioned miRNA inhibits VCAM1 expression, promoting tumor progression [117].

The ongoing search for both therapeutic and diagnostic targets in CC has allowed the identification of miR-210-3p, the expression of which was significantly higher in cervical cancer compared to a cervical intraepithelial neoplasia and normal cervical tissue. It turns out that miR-210-3p is closely related to the FIGO classification, lymphatic metastasis, and tumor differentiation. It was shown that miR-210-3p expression was higher in FIGO stage II/III than in stage I, high grade cancer was characterized by higher expression of miR-210-3p compared to medium and low-grade cancer, and patients with metastases to lymph nodes showed higher expression of the discussed miRNA [118].

The MiR-497-5p plays an important role in CC progression. It is the target of LINC00511, a long non-coding RNA, that induces invasion, migration, and proliferation of CC cells. The miR-497-5p counteracted the pro-cancer effect of LINC00511. The target of miR-497-5p is the MAPK1 (Mitogen-activated protein kinase 1) kinase, which promotes the development of cervical cancer cells. Indirectly, LINC00511 regulates MAPK1 by modulating miR-497-5p. Overexpression of miR-497-5p and downregulation of LINC00511 resulted in downregulation of the *MAPK1* gene. On the contrary, downregulation of miR-497-5p expression and the increase in LINC00511 expression induced *MAPK1* expression. Thus, the LINC00511/miR-497-5p/MAPK1 axis affects the progression of cervical cancer [119]. The prognostic function may also be fulfilled by miR-449a, that shows reduced expression in cervical cancer, induces cell proliferation, migration, and invasion. The MiR-449a shows a strong association with FIGO stage, differentiation stage, and lymph node metastases. It has been shown that higher expression of miR-499a is associated with longer survival time; on the contrary, low expression is associated with shorter survival time [120]. In Table 3, the miRNAs that are altered in expression in cervical cancer are shown.

## 6. Role of miRNAs in Cancer of the Vulva

Vulvar cancer is a rare type among gynecological cancers [124,125] and is classified according to the FIGO classification [124]. The pathogenesis of the most common type—squamous cell carcinoma of the vulva (SCC)—is complex and connected with HPV infection in which the suppressor proteins Rb and p53 are inactivated by the oncoproteins E6 and E7, and p16ink4a is overexpressed. In HPV-negative SCC, only p53 is overexpressed [126,127,128]. In another common type—vulvar melanoma—mutations in the KIT gene, and less frequently in NRAS and BRAF, are present [126,129,130,131]. Basal cell carcinoma can arise from chronic irritation, immunosuppression, trauma, or pelvic radiation [126,132]. In addition, it may develop in Paget’s disease [126,133,134], nevoid basal cell carcinoma syndrome [126,135], and lichen sclerosus [126,133,136]. Paget’s disease may originate primarily from mammary gland cells or, according to another theory, originates from Toker cells [126,137,138].

The studies carried out so far prove the significant roles of miRNAs in this type of gynecological cancer. One of them is miR-3147, which is overexpressed in vulvar squamous cell carcinoma. Increased expression induces invasion, migration, progression from G1 to S, and tumor cell proliferation. Moreover, it is related to the depth of cell invasion. The MiR-3147 also regulates the expression of the gene *SMAD4* (Mothers against decapentaplegic homolog 4) [139]. De Melo Maia et al. [140] showed that miR-223-5p is down-regulated in vulvar cancer cell lines. As a result of increased expression of miR-223-5p, reduced migration and proliferation of vulvar cancer cells was observed. Increased cellular invasion was also observed. In addition, the target of miR-223-5p has been shown to be Tumor protein p63 (TP63) [140]. Decreased expression of miR-103a-3p, miR-603 and miR-107 was found, as well as increased expression of miR-182-5p, miR-183-5p, and miR590-5p. A link between lymph node metastases and miR-590-5p has been demonstrated. Regarding miR-590-5p, it has been shown that its increased expression affects the reduction of TGFβRII, G1-S transformation, and increased cell migration and proliferation [141]. Overexpression of miR-4712-5p affects invasion and cell proliferation in vulvar squamous cell carcinoma. In addition, it targets PTEN, reducing its expression, and may affect the PTEN/AKT/p-GSK3β/cyclin D1 signaling pathway, affecting cell proliferation [142].

One of the genes activated in squamous cell carcinoma of the vulva is *HMGA2* (High mobility group AT-hook 2 gene), promoting cell proliferation, metastases, and epithelial-mesenchymal transition [143,144,145]. It seems that the *HMGA2* expression may be influenced by miRNAs. It has been shown that miR-30c and let-7a affect the regulation of this gene in the SSC of the vulva. Significant reductions in the expression of miR-30c and let-7a were found in vulvar SCC. It can therefore be concluded that *HMGA2* in the discussed type of gynecological cancer may be affected by miRNA dysregulation. It was also found, together with the overexpression of *HMGA2*, a decrease in the expression of the suppressor gene *FHIT* (Fragile Histidine Triad) [143]. The FHIT protein plays a significant role in the epithelial-mesenchymal transition and apoptosis [143,146,147].

A comparison of miRNA expression in vulvar cancer was also made between HPV-negative and HPV-positive tumors. Molecules with lower expression, with the presence of HPV, found in vulvar cancer include hsa-miR-1291, hsa-miR-193a-5p, hsa-miR-342-3p, hsa-miR-106b-3p, hsa-miR-365a-3p, hsa-miR-22-5p, hsa-miR-144-5p, hsa-miR-151a-5p, hsa-miR-519b-3p, hsa-miR-125b-1-3p, hsa-miR-19b-1-5p, hsa-miR-26b-3p, hsa-miR-1254-1, RNU44, and rno-miR-29c-5p. Molecules with higher expression, with the presence of HPV, include hsa-miR-142-3p, hsa-miR-1274b, hsa-miR-708-5p, hsa-miR-21-5p, hsa-miR-660-5p, hsa-miR-16-5p, hsa-miR-1267, hsa-miR-29c-5p, hsa-miR-186-5p, and hsa-miR-454-3p [148]. Table 4 lists the miRNAs that are altered in expression in vulvar cancer.

## 7. MicroRNAs as Potential Biomarkers in Gynecological Cancers

Biomarkers are potential indicators that provide important information about the pathogenesis, progression, and prognosis of many diseases, including the gynecological cancers discussed above. A commonly recognized marker is cancer antigen 125 (CA 125), which is a glycoprotein encoded by the *MUC16* gene. Currently, it is the most frequently determined serum marker in the diagnosis of ovarian cancer [149,150]. However, its specificity is quite low due to the relatively frequent occurrence of elevated CA-125 levels in women with endometriosis, benign ovarian cysts, myomas, as well as with non-gynecological malignancies or heart failure [151,152,153,154,155]. Another important marker is the carcinoembryonic antigen (CEA) used in the diagnosis of ovarian cancer; however, it is also used in the diagnosis of gastrointestinal cancers [156,157]. Human epididymis protein 4 (HE4), encoded by the gene *WFDC2* (WAP Four-Disulfide Core Domain 2), turns out to be an important marker not only in diagnosis but also in prognosis, monitoring and as a prognostic factor in endometrial cancer [158]. Due to the diagnostic non-specificity of the above-mentioned markers, research is still underway on new, more sensitive, and specific markers that enable rapid diagnosis, monitoring, and prediction. It transpires that the miRNAs discussed in this paper can be potential biomarkers that increase diagnostic efficiency in combination with commonly used markers. Su et al. [159] showed that the combination of the overexpressed miRNAs, miR-1307 and miR-375, and CA-125 and HE4 significantly increases the diagnostic accuracy of ovarian cancer [159]. Oliveira et al. [68], combining determinations of miR-200c-3p and miR-221-3p expression together with CA-125 obtained significant diagnostic accuracy (AUC = 0.96) in ovarian cancer [68]. Zhu et al. [160] also confirmed that the determination of CA-125 and HE4 together with miRNA-205 significantly increased the AUC (0.951), specificity (86.1%), and diagnostic sensitivity (100%) [160]. Yokoi et al. [57] showed that the combination of CA-125 with miR200a-3p, miR-26a-5p, miR-766-3p, let-7d-5p, miR-142-3p, and miR-328-3p resulted in diagnostic accuracy values of AUC (0.994), sensitivity (0.984), and specificity (0.956) [57].

It is worth noting that some of the miRNAs presented in this paper are identically altered in expression in more than one gynecological neoplasm. The following are downregulated: miR-152 (ovarian, endometrial, and cervical cancer), miR-214 (ovarian and cervical cancer), let-7a (ovarian and vulvar cancer), miR-145 (ovarian and cervical cancer), and miR-30c (endometrial and vulvar cancer). The following molecules are upregulated: miR-93 (ovarian and cervical cancer), miR-200a (ovarian, endometrial and cervical cancer), miR-200b (ovarian and cervical cancer), miR-200c (ovarian, endometrial and cervical cancer), miR-141 (ovarian and cervical cancer), miR-429 (ovarian and cervical cancer), miR-182 (ovarian and endometrial cancer), miR-182-5p (ovarian and vulvar cancer), miR-21-5p (ovarian and vulvar cancer), miR-205 (ovarian and endometrial cancer), miR-210 (endometrial and cervical cancer), and miR-183-5p (ovarian and vulvar cancer). In addition, some miRNAs show reduced or increased expression depending on the type of cancer, including miR-31 (up-regulated in cervical cancer, down-regulated in ovarian cancer), miR-142-3p (up-regulated in vulvar cancer, down-regulated in ovarian cancer), miR-203 (up-regulated in ovarian and endometrial cancer, down-regulated and cervical cancer), miR-221-3p (up-regulated in ovarian cancer, down-regulated in endometrial cancer), and miR-107 (up-regulated in endometrial cancer, down-regulated in vulvar cancer). Determination of specific miRNAs, both expressed in the same way in several gynecological neoplasms, as well as molecules whose expression differs between cancers, affects the possibility of developing a set of potential biomarkers that increase the sensitivity and diagnostic specificity of traditionally used markers. The above data is presented in Table 5.

In addition to their important function in the diagnostic scheme of gynecological cancers, miRNAs may also influence the regulation of resistance to therapy of these cancers. Examples are the PARP (poly-ADP ribose polymerase) inhibitors used in the treatment of ovarian cancer. It has been shown that the miR-200c molecule significantly affects the anticancer efficacy of treatment with PARP inhibitors [161]. Furthermore, miR-506-3p has a positive effect on the response to PARP inhibitors in ovarian cancer [162]. This proves the potential possibility of using miRNA molecules in the optimization of the therapeutic regimen as well as the possibility of understanding the mechanism of chemoresistance [161].

## 8. Conclusions

Gynecological cancers are the most common types of cancer among women. The ambiguity of the pathogenesis and diagnostic difficulties result in diagnosis of these diseases that is often too late. MicroRNAs provide a hope for the development of new, sensitive, and specific biomarkers enabling a better understanding of the molecular pathogenesis of gynecological cancers. To date, many detailed studies have been carried out to identify and characterize miRNAs, which are not only key diagnostic indicators, but also prognostic and predictive.

## Figures and Tables

**Table 1 biomedicines-11-01704-t001:** MicroRNAs with altered expression in ovarian cancer.

Down-Regulated miRNAs	Up-Regulated miRNAs	
miR-152 miR-214 miR-145 let-7a let-7c let-7b	miR-93 miR-200a miR-325 miR-200c miR-200b miR-141 miR-492 miR-429 miR-182	Braicu et al. [51]
hsa-miR-1271-5p hsa-miR-574-3p	hsa-miR-182-5p hsa-miR-96-5p hsa-miR-183-5p hsa-miR-182-3p hsa-miR-15b-5p hsa-miR-141-5p hsa-miR-135b-3p hsa-miR-130b-5p	Wang et al. [61]
miR-214 miR-31	miR-155	Mitra et al. [64]
	miR-141 miR-21 miR200b miR-200a miR-200c miR-205 miR-203 miR-214	Taylor et al. [66]
	miR-99a-5p	Yoshimura et al. [67]
miR-195-5p miR-451a	miR-200c-3p miR-21-5p miR-221-3p miR-484	Oliveira et al. [68]
miR-142-3p		Gao et al. [69]

**Table 2 biomedicines-11-01704-t002:** MicroRNAs found in endometrial cancer that are potential biomarkers.

Down-Regulated miRNAs	Up-Regulated miRNAs	
miR-23b miR-199a-3p miR-125b-5p miR-451 miR-221-3p	miR-34a-5p miR-146-5p	Klicka et al. [89]
miR-let-7i miR-221 miR-152 miR-193 miR-30c	miR-185 miR-106a miR-181a miR-423 miR-210 miR-103 miR-let7c miR-107	Boren et al. [102]
	miR-95 miR-210 miR-200c miR-203 miR-103 miR-182 miR-151 miR-200a miR-183 miR-155 miR-194 miR-205 miR-223 miR-106a	Chung et al. [99]
miR-204	miR-186 miR-223 miR-222	Montagnana et al. [103]
miR-383-5p miR-34c-5p miR-34c-3p miR-10b-5p miR-449b-5p miR-2110 miR-200b-3p miR-34b-3p		Roman-Canal et al. [101]

**Table 3 biomedicines-11-01704-t003:** MicroRNAs that are altered in expression in cervical cancer.

Down-Regulated miRNAs	Up-Regulated miRNAs	
miR-599		Gong et al. [112]
miR-489		Juan et al. [113]
miR-1298-3p		Li et al. [114]
	miR-210-3p	Shao et al. [118]
miR-497-5p		Lu et al. [119]
miR-449a		Wang et al. [120]
miR-203 miR-193b	miR-518a miR-34c miR-34b miR-20b miR-9 miR-338 miR-424 miR-512-5p miR-10a miR-345	Cheung et al. [121]
miR-143 miR-26a miR-145 miR-203 miR-99a miR-513 miR-199a miR-29a	miR-148a miR-10a miR-196a miR-132 miR-302b	Pereira et al. [122]
hsa-miR-127 hsa-miR-376a hsa-miR-218 hsa-miR-214 hsa-miR-1 hsa-miR-145 hsa-miR-368 hsa-miR-99a hsa-miR-100 hsa-miR-320 hsa-miR-195 hsa-miR-497 hsa-miR-152 hsa-miR-99b hsa-miR-143 hsa-miR-10b rno-miR-140 mmu-miR-140 rno-miR-10b	hsa-miR-7 hsa-miR-141 hsa-miR-429 hsa-miR-31 hsa-miR-142-5p hsa-miR-200a hsa-miR-20b hsa-miR-224 hsa-miR-200b hsa-miR-18a hsa-miR-146b hsa-miR-93 hsa-miR-200c hsa-miR-210 hsa-miR-20a rno-miR-31 rno-miR-93 PREDICTED_MIR189	Rao et al. [123]

**Table 4 biomedicines-11-01704-t004:** MicroRNAs that are altered in expression in vulvar cancer.

Down-Regulated miRNAs	Up-Regulated miRNAs	
	miR-3147	Yang et al. [139]
miR-223-5p		De Melo Maia et al. [140]
miR-103a-3p miR-603 miR-107	miR-182-5p miR-183-5p miR-590-5p	Yang et al. [141]
	miR-4712-5p	Yang et al. [142]
miR-30c let-7a		Agostini et al. [143]
hsa-miR-1291 hsa-miR-193a-5p hsa-miR-342-3p hsa-miR-106b-3p hsa-miR-365a-3p hsa-miR-22-5p hsa-miR-144-5p hsa-miR-151a-5p hsa-miR-519b-3p hsa-miR-125b-1-3p hsa-miR-19b-1-5p hsa-miR-26b-3p hsa-miR-1254-1 RNU44 rno-miR-29c-5p	hsa-miR-142-3p hsa-miR-1274b hsa-miR-708-5p hsa-miR-21-5p hsa-miR-660-5p hsa-miR-16-5p hsa-miR-1267 hsa-miR-29c-5p hsa-miR-186-5p hsa-miR-454-3p	De Melo Maia et al. [148]

**Table 5 biomedicines-11-01704-t005:** MicroRNAs that are involved in gynecological cancer.

microRNA	Down-Regulated	Up-Regulated
miR-152	OC EC CC	
miR-214	OC CC	
Let-7a	OC VC	
miR-145	OC CC	
miR-30c	EC VC	
miR-93		OC CC
miR-200a		OC EC CC
miR-200b		OC CC
miR-200c		OC EC CC
miR-141		OC CC
miR-429		OC CC
miR-182		OC EC
miR-182-5p		OC VC
miR-21-5p		OC VC
miR-205		OC EC
miR-210		EC CC
miR-183-5p		OC VC
miR-31	OC	CC
miR-142-3p	OC	VC
miR-203	CC	OC EC
miR-221-3p	EC	OC
miR-107	VC	EC

OC—ovarian cancer; EC—endometrial cancer; CC—cervical cancer; VC—vulvar cancer.

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
