# Peer review of "MicroRNAs as Potential Biomarkers in Gynecological Cancers"

_biomedicines, 2023, doi:10.3390/biomedicines11061704_

Round 1
Reviewer 1 Report
MicroRNAs review about gynecologic cancers are recommendable but needs revision.
1. Detailed explanation about predictive and prognostic biomarker nature of miRNA are necessary: new, sensitive, and specific biomarkers, key diagnostic indicators. Among table only “up and down regulation” are indicated. How does author compare characteristics of miRNA among gynecologic cancer? Is it better than tumor markers (CA 125, SCC, CEA) or imaging studies (CT, MRI) or target chemo treatment like standard PARP inhibitor or Immune check point inhibitor or any connection with miRNA?
2. Detailed explanation of gynecologic disease looks not necessary in the manuscript. (3.1. Ovarian cancer,4.1. Uterine cancer, 5.1. Cervical cancer, 6.1. Vulvar cancer).
MicroRNAs review about gynecologic cancers are recommendable but needs revision.
1. Detailed explanation about predictive and prognostic biomarker nature of miRNA are necessary: new, sensitive, and specific biomarkers, key diagnostic indicators. Among table only “up and down regulation” are indicated. How does author compare characteristics of miRNA among gynecologic cancer? Is it better than tumor markers (CA 125, SCC, CEA) or imaging studies (CT, MRI) or target chemo treatment like standard PARP inhibitor or Immune check point inhibitor or any connection with miRNA?
2. Detailed explanation of gynecologic disease looks not necessary in the manuscript. (3.1. Ovarian cancer,4.1. Uterine cancer, 5.1. Cervical cancer, 6.1. Vulvar cancer).
Reviewer 2 Report
The authors systematically summarize the changes in microRNA expression levels in gynecological tumors by tumor. I hope that this will be a useful review when considering biomarker development.
minor comment: In Table 2, the expression status of the mir200 family is slightly different between Ref. 97 and Ref. 99. Both mir-200a and mir-200b are matured from a single transcript, so why do these differences in expression appear? Please discuss.
Reviewer 3 Report
Miśkiewicz presented a review article aimed at collecting all the miRNAs involved in the pathogenesis of gynecological tumors or as potential biomarkers. Overall, the structure of the review is very simple and no in-depth literature analyses were performed (e.g. no evaluation of the genes targeted by the miRNAs listed in the different tables or specific molecular mechanisms driven by the selected miRNAs). Nevertheless, the manuscript can be interesting for the researchers involved in this field of investigation. Below are reported some minor/major comments that will improve the manuscript:
1) Please provide supporting references for the following sentences: “Gynecological cancers are the most common cancers in women. There may be e.g. in the ovaries, endometrium, cervix and vulva. The still ambiguous pathogenesis, heterogeneity of diseases and difficult diagnostics mean that gynecological cancers are often diagnosed at a late stage. Therefore, research is still underway on new, sensitive and specific biomarkers enabling quick, non-invasive diagnostics in the early stages of the disease, prediction and precise monitoring of therapy.”. For this purpose, please see:
- PMID: 35282075
- PMID: 34132354
- PMID: 33327492
- PMID: 33994735
2) In the following sentence: “However, the genetic factor is indicated as the main cause of the disease, especially BRCA1 and BRCA2 gene mutations [36,37].”, please note that Refs 36 and 37 do not describe in detail the involvement of BRCA1/2 mutations in ovarian cancer. Please consider to substitute these references with more updated ones. For this purpose, please see:
- PMID: 35267543
- PMID: 35383859
- PMID: 28794804
- PMID: 31467961
3) Report the FIGO Classification of ovarian cancer in a tabular format (add a new table);
4) Avoid the use of the expression “miRNA molecule” or the name of specific miRNAs followed by the word “molecules”;
5) For each cancer type described, I strongly suggest to summarize the clinical and pathological features (e.g. avoid the description of staging or other non-relevant information);
6) Before the conclusions, the authors should present a paragraph describing miRNAs shared among the different gynecological cancers with a new table reporting the miRNAs that are concordantly up-regulated or down-regulated in the different tumors. Such data are necessary in order to propose novel potential diagnostic or prognostic biomarkers for the management of gynecological tumors.
Round 2
Reviewer 1 Report
I suggest 3.1. Ovarian cancer,4.1. Uterine cancer, 5.1. Cervical cancer, 6.1. Vulvar cancer are all unnecessary regarding this subject.
I suggest 3.1. Ovarian cancer,4.1. Uterine cancer, 5.1. Cervical cancer, 6.1. Vulvar cancer are all unnecessary regarding this subject.
Author Response
Dear Reviewer,
Thank you for your constructive comment.
Comment:
I suggest 3.1. Ovarian cancer,4.1. Uterine cancer, 5.1. Cervical cancer, 6.1. Vulvar cancer are all unnecessary regarding this subject.
Answer:
We have removed the subchapters and renamed the chapters on the role of miRNAs in individual cancers.
Reviewer 3 Report
Dear Authors,
all of my previous comments were properly addressed. The manuscript can be accepted for publication after the editorial check.
Author Response
Dear Reviewer,
Thank you for your constructive comments. The suggested changes were necessary to improve the manuscript.
Round 3
Reviewer 1 Report
Well corrected thanks
Well corrected thanks